# Optimization of Water Distribution Networks Using Genetic Algorithm Based SOP–WDN Program

**Uchit Sangroula** [1], **Kuk-Heon Han** [2], **Kang-Min Koo** [1], **Kapil Gnawali** [1] **and Kyung-Taek Yum** [1,*]

1   Graduate School of Water Resources, Sungkyunkwan University, Suwon 16419, Korea; uchit@skku.edu (U.S.); koo00v@skku.edu (K.-M.K.); kapil@g.skku.edu (K.G.)
2   Smart Water Technology and Consulting, Co., Ltd., Incheon 21315, Korea; kuk0904@daum.net
*   Correspondence: kwfyum@skku.edu; Tel.: +82-31-290-7645

**Abstract:** Water distribution networks are vital hydraulic infrastructures, essential for providing consumers with sufficient water of appropriate quality. The cost of construction, operation, and maintenance of such networks is extremely large. The problem of optimization of a water distribution network is governed by the type of water distribution network and the size of pipelines placed in the distribution network. This problem of optimal diameter allocation of pipes in a distribution network has been heavily researched over the past few decades. This study describes the development of an algorithm, 'Smart Optimization Program for Water Distribution Networks' (SOP–WDN), which applies genetic algorithm to the problem of the least-cost design of water distribution networks. SOP–WDN demonstrates the application of an evolutionary optimization technique, i.e., genetic algorithm, linked with a hydraulic simulation solver EPANET, for the optimal design of water distribution networks. The developed algorithm was applied to three benchmark water distribution network optimization problems and produced consistently good results. SOP–WDN can be utilized as a tool for guiding engineers during the design and rehabilitation of water distribution pipelines.

**Keywords:** water distribution networks; optimization; genetic algorithm; EPANET

## 1. Introduction

A water distribution network (WDN) is comprised of various elements, such as reservoirs, pumps, pipes, tanks, and valves. Around 80% of the total cost of a water supply project is invested in its water distribution system [1]. Hence, the design of a cost-effective and reliable water distribution network is a must. Optimization of the WDN involves the design of a reliable, efficient, and cost-effective distribution network that fulfils the necessary water demands, while maintaining adequate pressure heads.

This is crucial for conservation of water resources, as well as for reducing energy requirements and maintenance costs. Optimization of WDNs can be categorized into many types, viz. design, operation, calibration, level-of-service, monitoring system and network testing. This paper deals with determining the optimal diameters of pipelines in a water distribution network with a predetermined layout.

*Background and Related Work*

Over the years, numerous researchers have presented many different methods for obtaining the optimal solution to the pipe network optimization problem. The Hardy cross method is considered as the oldest method for solving a pipe network. In this method, at any pipe junction, the algebraic sum of flow must be zero, and the algebraic sum of pressure drops at any loop must also be zero [2]. This method was improved upon by many other researchers. Alperovits and Shamir [3] proposed one of the most significant approaches for solving the problem of water distribution network design by utilizing the successive Linear Programming Gradient (LPG) method. This method was adopted and further expanded upon by other researchers [4,5].

However, deterministic methods, such as linear programming and non-linear programming, presented drawbacks, such as entrapment in local minima, and dependence on the starting point. Hence, they failed to obtain near optimal solutions for complex, multi-objective, real-world pipe network problems. It is crucial to escape local minima [6] and to overcome these drawbacks, researchers began to utilize meta-heuristic algorithms (genetic algorithms, simulated annealing, etc.) for water network design problems. These techniques include algorithms having some stochastic components. Goldberg and Kuo introduced stochastic methods for the optimization of water distribution networks using the principles of natural selection and genetics [7]. Simpson et al. used simple genetic algorithms (GA), and obtained a near optimal solution [8], while Simpson et al. [9] compared the GA technique with other methods, such as complete enumeration and non-linear optimization, and concluded that the GA technique generates multiple alternative solutions that are both practical and close to the optimum. The results obtained by Simpson et al. [8] were further improved upon by Dandy et al. [10] using the concept of variable power scaling of the fitness function, an adjacency mutation operator, and gray codes. Savic and Walters developed the computer model GANET [11] that utilizes GA for the least-cost design of pipe networks.

To avoid unfeasible solutions due to the violation of constraints, a penalty factor is necessary during the selection process of GA. Deb and Agrawal [12] developed a niched-penalty method to more effectively solve constrained optimization problems using GAs. Wu and Simpson [13] demonstrated significant improvements in efficiency and robustness for single-objective optimization utilizing a boundary search method. Liong and Atiquzzaman used the shuffled complex evolution (SCE) linked with EPANET hydraulic network solver [14] to obtain the least cost of some well-known water distribution networks in the literature. SCE was demonstrated to be a potential alternative to other optimization algorithms, due to its faster computational speed. Other algorithms, such the shuffled frog-leaping algorithm (SFLA) by Eusuff [15] and the harmony search Algorithm (HS) by Geem [16], have obtained comparable results, and have proven to be effective tools for the optimal design of water networks.

Some studies consider a single economic objective (least-cost) to formulate the network optimization and rehabilitation problem, whereas others consider a multi-objective optimization approach that compares interesting trade-offs (e.g., a slight pressure deficit can sometimes be outweighed by substantial cost reduction) [17]. To improve network reliability, Chandramouli and Malleswararao [18] used fuzzy logic based on the excess pressure available at demand nodes. Jin et al. analyzed additional objectives, such as considering both pressure and velocity violations [19]. Prasad and Park utilized genetic algorithms and considered both minimization of cost and maximization of network reliability [20]. More recent developments include improving algorithm convergence by using an engineered initial population, rather than a random one [21]; improvement of computational efficiency via the reduction of search space [22]; combining GA and mathematical programming with the inclusion of new elements such as pressure reducing valves [23]; using artificial neural networks (ANNs) rather than hydraulic and water quality simulation models together with differential evolution (DE) for optimization [24]; developing the Harris hawks optimization algorithm (HHO) for WDN optimization [25].

Additionally, Bilal and Pant utilized a hybrid metaheuristic algorithm (FA-PSO) [26] focusing on the Hanoi distribution network. Praneeth et al. demonstrated water cycle optimization algorithm [27], whereas Pankaj et al. utilized Cuckoo search [28] for least cost design of benchmark distribution networks. Surco et al. utilized a modified particle swarm optimization (PSO) algorithm for the optimization of distribution networks [29]. Cassiolato et al. proposed a deterministic mathematical programming approach without utilizing hydraulic simulators for cost minimization of looped WDNs, where generalized disjunctive programming is used to reformulate the discrete optimization problem to a mixed-integer nonlinear programming (MINLP) problem [30]. Pant and Snasel proposed a fuzzy C-means adaptive differential evolution (FCADE) for optimizing well-known

benchmark WDN problems [31]. Bi et al. compared the searching behavior of evolutionary algorithms on water distribution system design optimization [32]. Zhao et al. proposed an in-sync optimization model for network layout and pipe diameter determination of a self-pressurized drip irrigation system [33]. Shao et al. utilized genetic algorithm for optimal placement of flow meters and valves in a distribution system [34].

Moreover, many researchers have considered operational optimization of WDNs, since minimization of operational energy costs during pumping must also be accounted for together with the construction costs. Electricity consumption during operation is one of the biggest marginal expenditures for water utilities [35]. Operational optimization can be achieved by controlling times when pumps operate, also called pump scheduling [36,37], flow rates [38], pump speeds [39] and tank-water trigger levels [40]. Furthermore, other factors such as real-time control and water quality can also be considered [41–43]. Metaheuristics such as GA have been abundantly utilized for operational optimization of WDNs, primarily with the objective of minimizing the overall cost of the pumping operation [44–46]. Research works in optimization of WDNs have been comprehensively reviewed by Mala et al. for design and rehabilitation [47] and system operation [35]. GA have also been abundantly applied for optimization in multiple fields of research [48–50]. Katoch et al. [51] have elaborated the advancements made in the field of GA.

## 2. Materials and Methods

### 2.1. Problem Formulation

Cost-effective WDN design is a discrete optimization problem, as the individual pipe sizes are to be selected from a list of available commercial size diameters. The search space can be determined as the number of available diameters, raised to the power of the number of pipes in the network [52]; e.g., if 8 different commercial pipe sizes are available for the design of a WDN having 10 pipelines, the search space size would be $8^{10}$, i.e., 1,073,741,824 different pipe combinations. Hence, even for a relatively small pipe network, the search space is large. The design of an economically optimal water distribution network is a difficult task, because it involves solving many complex, non-linear, and discontinuous hydraulic equations, while simultaneously optimizing pipe sizes and other network components [53,54].

Optimization of a water distribution network aims to find the optimal pipe diameters in the network for the given layout and demand requirements. The optimal pipe sizes that satisfy all implicit constraints (conservations of mass and energy), and explicit constraints (hydraulic and design constraints) are selected in the final network.

The continuity equation is given as:

$$\sum_{i=1}^{n} q_i = 0 \tag{1}$$

The continuity equation is applied to each node, with $q_i$ being the flow rate (flow into and flow out of the node), and $n$ is the number of pipes connected at the node.

The energy equation is given as:

$$\sum_{i=1}^{m} h_i = 0 \tag{2}$$

The energy equation is applied to each loop in the distribution network, where $h_i$ is the head loss in each pipe, and $m$ is the number of pipes in the loop.

The objective function is the total cost of the given network. The total cost $C_T$ is calculated as:

$$C_T = \sum_{i=1}^{N_p} C_i(D_i) \cdot L_i \tag{3}$$

where, $N_p$ is the total number of pipes, $C_i(D_i)$ is the cost per unit length of pipe *i* with diameter $D_i$, and $L_i$ is the length of pipe *i*. The objective function is to be minimized under the implicit constraints and explicit constraints.

The head loss is the sum of the local head losses and the friction head losses. The equation used to calculate the head loss is the Hazen–Williams equation. This equation is an empirical equation that relates the flow of water in a pipe with the physical properties of the pipe and the energy loss due to friction. The Hazen–Williams coefficient, abbreviated as *C*, is a dimensionless number used in the Hazen–Williams equation [55]. The equation can be expressed as:

$$h_f = 4.72 C^{-1.85} Q^{1.85} D^{-4.87} L \tag{4}$$

where, $h_f$ is the head loss, *Q* is the flow rate, *C* is the Hazen–Williams coefficient, *D* is the pipe inside diameter, and *L* is the pipe length.

*2.2. Genetic Algorithm*

A genetic algorithm (GA) is a search algorithm based on the mechanics of natural selection and natural genetics [56]. Although stochastic at certain aspects, a genetic algorithm is not entirely random, as it utilizes historical information to determine new search points. GAs have been widely utilized to solve optimization problems in multiple fields [57]. Following the concept of 'survival of the fittest', improvements in solutions evolve from past generations, until a near optimal solution is obtained. In genetic algorithms, the candidate solutions are represented by chromosomes (e.g., binary strings), and are collectively known as the population. The chromosomes are then evolved in each subsequent generation, according to their fitness. The fitness evaluation of each candidate solution depends upon how well it the meets the requirements of a pre-defined objective function (e.g., lowest cost). The more fit the candidate solution, the greater probability it will have of being selected for reproduction. Hence, the more fit chromosomes replace the less fit chromosomes, and the process continues until a near optimal solution is found.

The general idea of GA in a pipe network optimization problem is to select a population of initial solution points, scattered randomly in the optimization space, and then converge iteratively to better solutions, until the desired criteria for stopping are achieved. The steps for using GA for pipe network optimization can be briefly described as follows [8]:

1. Generation of initial population

The GA randomly generates an initial population of coded strings (binary) representing pipe network solutions of population size N. Each of the N strings represents a possible combination of pipe sizes.

2. Computation of network cost

For each N string in the population, the GA decodes each substring into the corresponding pipe size and computes the total network cost (material cost, construction cost, etc.).

3 Hydraulic analysis of each network

A steady state hydraulic network solver computes the heads and discharges under the specified demand patterns for each of the network designs in the population. The actual nodal pressures are compared with the minimum allowable pressure heads, and any pressure deficits are noted. Similarly, the actual water velocities at pipes are compared with the desired velocity of the water distribution network and any deviation in velocity are also noted.

4. Computation of penalty cost

The GA assigns a penalty cost for each individual network design in the population if a pipe network does not satisfy the pressure and velocity constraints (for example, pressure violation at a particular node if the pressure in the node is less than or greater than the desired pressure).

5. Computation of total network cost

The total cost of each network in the current population is then taken as the sum of the network cost (Step 2) and the penalty cost (Step 4).

6. Computation of the fitness

The fitness of the coded string is taken as some function of the total network cost. For each proposed pipe network in the current population, fitness can be computed as the inverse or the negative value of the total network cost (Step 5).

7. Generation of a new population using the selection operator

The GA generates new members for the next generation by a selection scheme that depends on the fitness of the initial members.

8. The crossover operator

Crossover occurs with some specified probability for each pair of parent strings selected in Step 7. A uniform type of crossover operator is commonly used to accompany the comparatively large string size for pipe network optimization.

9. The mutation operator

Mutation occurs with some specified probability of mutation for each bit in the strings that have undergone crossover. The purpose of the mutation operator is to maintain genetic diversity from one generation of a population to another.

10. Production of successive generations

The use of the three operators described above produces a new generation of pipe network designs using Steps 2 to 9. The GA repeats the process to generate successive generations. The final costs and pipe network designs are stored, and the cheaper cost alternatives that meet the required constraints are updated.

*2.3. EPANET*

EPANET is a hydraulic simulator that can perform extended hydraulic and water quality simulations for pressurized water distribution networks (Rossman, [58]). Generally, a water distribution network consists of many elements, such as pipes (links), pipe junctions (nodes), pumps, control valves, and tanks/reservoirs. EPANET solves the water distribution network for the flow of water in each pipe, pressure at each junction, water height in each tank, concentration of chemical species, etc. During the hydraulic analysis of the water distribution network, EPANET solves both the conservation of mass and energy equations.

EPANET–MATLAB Toolkit is a software for interfacing a drinking water distribution system simulation library, EPANET, with the MATLAB technical computing language developed by Eliades [59]. The Toolkit allows users to access EPANET and EPANET-MSX through their shared object libraries, as well as their executables. EPANET can be called and used through a programming interface by an external software, which can be written in different programming languages (such as C/C++, Python, or MATLAB). Generally, a large number of commands have to be written to achieve specific results, such as extracting the node pressures, pipe diameters, pipe roughness coefficients, or specifying demand patterns. However, in the EPANET–MATLAB toolkit, a significant part of the repetitive code is already included in the toolkit functions and can be used directly.

*2.4. SOP–WDN*

Smart optimization program for water distribution networks (SOP–WDN) is an algorithm that has been developed by Smart Water Grid (SWG) research works for water distribution network optimization by using a genetic algorithm. The algorithm is written in MATLAB programming language. It interfaces with EPANET-MATLAB toolkit [59] (hydraulic solver integrated within SOP-WDN) for steady state hydraulic simulation and

solution. The algorithm imports the network layout and data from EPANET. Design parameters, such as available pipe sizes, respective cost of pipes, roughness coefficient, and required pressure and velocities for the network, are to be added. GA optimization parameters, such as population size, crossover probability, and mutation rate, are prerequisites for the algorithm, and can be set by the user. Figure 1 shows a flowchart of the overall algorithm:

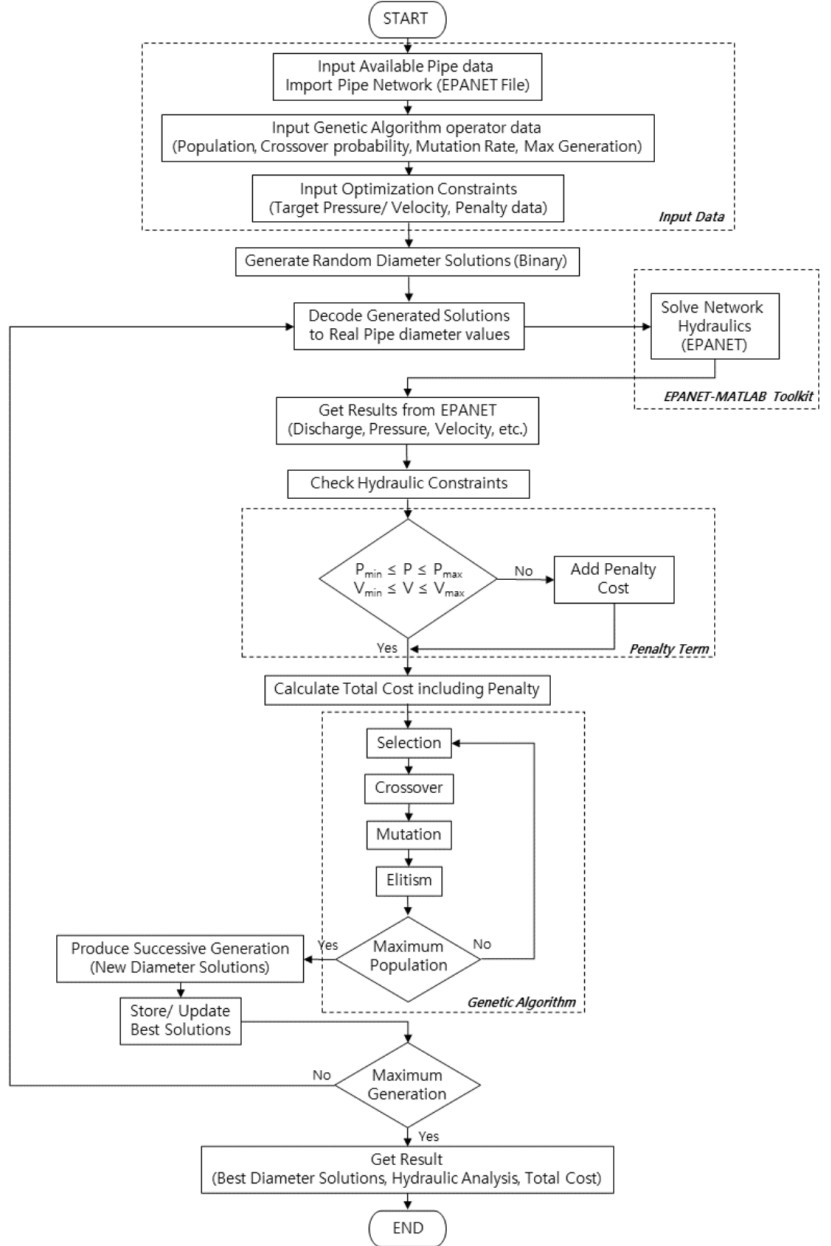

**Figure 1.** General Flowchart of the SOP–WDN.

### 2.4.1. Encoding Scheme, Interpretation and Redundancy

A sequence of binary numbers is used to represent a water distribution network in SOP–WDN, as it enables a relatively extensive exploration of the search space. The coding scheme is also simple in implementation for the given task. Reflected binary code (RBC) or grey ode is utilized since it assures that two successive values differ by only one bit (binary digit). The number of bits required to represent an individual pipe in the network depends upon the number of available pipe diameters. The total binary sequence represents the

entire pipe network. An example for the process of interpretation of a water distribution network by the algorithm is demonstrated.

Consider, a water distribution network having 1 reservoir, 1 tank, 5 demand nodes and 6 pipelines (all 100 m in length); there are 8 available pipe sizes for this network as shown in Table 1.

**Table 1.** Example: Available pipe sizes.

| S.N. | Available Pipe Sizes (mm) | Unit Cost (per m) | 3-Bit (Binary) Representation | 3-Bit (Grey) Representation |
|------|---------------------------|-------------------|-------------------------------|------------------------------|
| 1 | 8 | 100 | 000 | 000 |
| 2 | 10 | 120 | 001 | 001 |
| 3 | 12 | 150 | 010 | 011 |
| 4 | 14 | 180 | 011 | 010 |
| 5 | 16 | 200 | 100 | 110 |
| 6 | 18 | 250 | 101 | 111 |
| 7 | 20 | 300 | 110 | 101 |
| 8 | 24 | 350 | 111 | 100 |

As there are 8 available pipe sizes in the example, the number of bits required to represent each individual pipe is 3 bits (starting from 000 to 100).

Any 18-character-long binary string can, hence, represent all 6 pipelines in this distribution network. Consider a randomly generated binary string of length 18 (exe. 101110011001000111). Since there are 6 pipes in the distribution system layout, the generated binary string of length 18 can be divided into six binary strings, each a length of 3 bits. Each binary string of 3 bits then represents a unique pipe size and the position of that binary string represents the position of the pipe in the overall network layout. The resulting pipe network and its respective cost can be obtained as shown in Table 2 and the network layout can be visualized as shown in Figure 2.

**Table 2.** Example: Interpretation of binary to pipe networks.

| Randomly Generated Binary String | 101110011001000111 | | | | | |
|----------------------------------|---------------------|---|---|---|---|---|
| **6 Individual Pipes** | **101, 110, 011, 001, 000, 111** | | | | | |
| **Pipe Position in WDN** | **Pipe No.1** | **Pipe No.2** | **Pipe No.3** | **Pipe No.4** | **Pipe No.5** | **Pipe No.6** |
| Chromosome (Binary) | 101 | 110 | 011 | 001 | 000 | 111 |
| Pipe Diameter (mm) | 20 | 16 | 12 | 10 | 8 | 18 |
| Unit Cost (per m) | 300 | 200 | 150 | 120 | 100 | 250 |
| Length (m) | 100 | 100 | 100 | 100 | 100 | 100 |
| Cost of individual pipe | 30,000 | 20,000 | 15,000 | 12,000 | 10,000 | 25,000 |
| Total Cost of Network | 112,000 | | | | | |

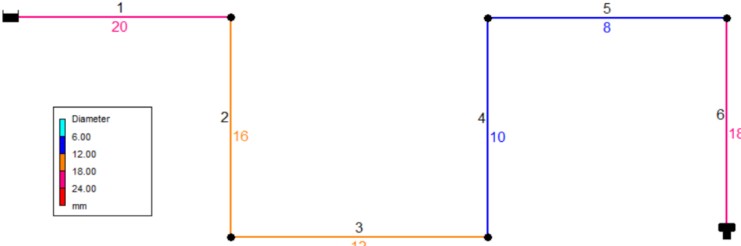

**Figure 2.** Example: Pipe network showing pipe position and diameter.

When a parameter belonging to a finite discrete set is encoded with binary notations, some of the codes may become redundant. For example, consider the design of a water

distribution system where the number of commercially available diameters is 13. In this case, a minimum of 4 bits are required for coding the elements of the set of diameters, and this gives $2^4$ = 16 discrete values. With 13 diameter options, three of the substrings become redundant (do not represent any diameters). The challenges of dealing with redundant binary codes have been overlooked in the literature and in published methods [60]. Saleh and Tanyimboh proposed representing redundant codes with closed pipes of fictitious diameters and low fitness values, assuming their extinction through evolution and natural selection [61]. However, this approach was found to prematurely lose valuable genetic information.

Alternatively, the redundant codes may be mapped to valid codes. In this case, some diameter options will be represented more than once. The mapping options may be fixed or random. In fixed mapping, the redundant codes are assigned to valid pipe diameter options prior to optimization. Random mapping assigns the redundant codes to the valid diameter options randomly. Tanyimboh [62] observed that, during mapping, over-representation of the largest pipe diameter performed better than over-representation of the smallest pipe diameter. It was also overserved that a balanced, unbiased allocation achieved good results. SOP-WDN handles redundant mapping by linearly scaling the probability of allocation of pipe diameters prior to optimization, where the largest pipe has 2 times the allocation chance than the smallest.

### 2.4.2. Genetic Algorithm Operators

Genetic algorithms differ from conventional optimization algorithms and search procedures as they work with a coding of the solution set and not the solutions themselves [56]. GA mainly consists of four basic operators: selection, crossover, mutation and elitism and utilizes these operators together to produce new generations.

1.  Selection

Selection is a crucial step in GA that determines whether a particular candidate will participate in the reproduction process or not. The selection operators give preference to the fitter chromosomes (candidate solutions), allowing them to pass on their 'genes' (information) to the next generation. Hence, worse chromosomes (with poor fitness) get eliminated. SOP-WDN algorithm computes fitness of an individual as the reciprocal of (the total cost times the imposed penalty). The selection operator then randomly picks chromosomes out of the population according to fitness. Roulette wheel, rank, and tournament selection are well-known techniques for selection and are commonly utilized in the literature.

In general, the probability of selection of an individual is calculated as the fitness values of the individual divided by fitness value of the population. Selection pressure is defined as the degree to which the better individuals are favored in the population; it drives the GA into improving the fitness over successive generations. In this study, power-law based probability selection was utilized, where $P(i)$ is the preserved probability based on rank of the $i$th ranked individual (ranked based on fitness) in the population a having total of n individuals. The value of $\tau = 1.1 \sim 1.2$ (controlling selection pressure) was found ideal after numerous simulations.

$$P(i) \ = \ \frac{i^{-\tau}}{\sum_{i=1}^{n} i^{-\tau}} \tag{5}$$

2.  Crossover

Crossover is a genetic operator that combines two chromosomes (parents) in order to produce new chromosomes (children). Some common crossover techniques utilized in the literature are one-point crossover, two-point or k-point crossover, uniform crossover, shuffle crossover, three-parent crossover. In one point crossover, a crossover point is selected at random through the length of the chromosome. Summation of genes from the first parent, put before crossover point and second parent after the crossover point creates the new child. In two-point or k-point crossover, genetic material is exchanged between two or

more random positions along the length of chromosome. Uniform crossover operates in individual genes of the selected chromosome, rather than on blocks.

Since GAs are problem specific, crossover is applied considering the chromosomes as group of distribution pipelines. Here, change of any single gene in a chromosome means changing and replacing the current pipe with another one of a different diameter. It is preferred to alter and test many different pipe combinations when searching for the optimal solution. After experimenting with varying crossover operators, k-point crossover was selected as the preferred crossover method. The number of crossover points is taken as $(0.8 \times N_p)$ rounded to the nearest integer, where $N_p$ is the total number of pipelines in the distribution system. The locations of crossover points are taken randomly. For example, the number of crossover points for a distribution network with 6 pipes is shown in Figure 3. The higher mixing ratio showed faster convergence. Uniform crossover also showed good results whereas the worst results were obtained from one-point crossover. The applied crossover probability was 85–90%.

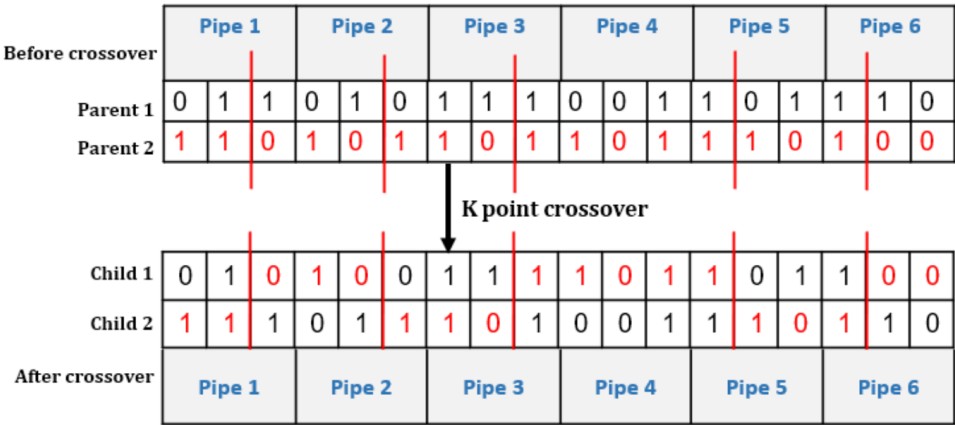

**Figure 3.** k-point crossover.

3. Mutation

After crossover, the chromosomes are subjected to mutation. Mutation prevents the algorithm from being trapped in a local optimum (premature convergence of the GA). It involves the modification of the value of each 'gene' of a chromosome with a small probability, called the mutation probability. Mutation plays the role of re-discovering lost or unexplored genetic materials, as well as for maintaining genetic diversity [56]. Mutation operators produce random changes in various chromosomes of the population.

Generally, a mutation probability of $(1/l)$, where $l$ is the chromosome length considered appropriate in the literature. In this study, mutation rate was kept around (4–6)%, which is considered higher, especially for networks with large number of pipelines, and a hill-climbing mutation operator was applied for the final 10% of the generations, which would permit mutation only if it improved the quality of the solution. The mutation strategy utilized in the study, shown in Figure 4, was bit-flip mutation since it complements the encoding scheme of the algorithm.

4. Elitism

Elitism involves selecting a small proportion of the fittest chromosomes of one generation and copying them directly into the next generation. Elitism is used to protect the fittest chromosomes from crossover and mutation. The least-cost distribution networks, without violating any pressure and velocity constrains per generation are considered as the elite chromosomes. The 2 best performing individuals from each generation are saved prior to crossover and mutation for the next generation.

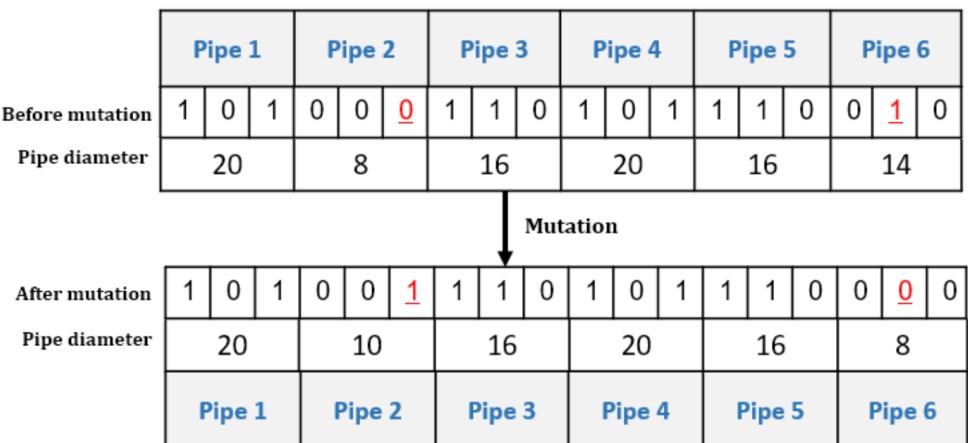

**Figure 4.** Bit flip mutation.

2.4.3. Penalty Functions

The following Equations (6) and (7) are used to calculate the penalty cost [63]. The penalty imposed will be higher, the larger the violations of pressure and velocity from the target values. The penalty equation for violation of pressure constraint in the water distribution network implemented by SOP–WDN can be represented as:

$$P_P = 1 + \sum_{j=1}^{N_n} |T_P - P_j| \cdot P_{P1} + \sum_{j=1}^{N_n} |T_P - P_j| \cdot P_{P2} \tag{6}$$

where, $N_n$ is the number of nodes in the network, $P_P$ the pressure penalty, $P_j$ is the pressure of node $j$, $T_P$ is the target pressure, $P_{P1}$ is the pressure penalty coefficient if the pressure at the node is above the target pressure, and $P_{P2}$ is the pressure penalty coefficient if the pressure at the node is below the target pressure.

The penalty equation for the violation of velocity constraint in the distribution network implemented by SOP–WDN can be represented as:

$$V_P = 1 + \sum_{i=1}^{N_p} |T_V - V_i| \cdot V_{P1} + \sum_{i=1}^{N_p} |T_V - V_i| \cdot V_{P2} \tag{7}$$

where, $N_p$ is the number of pipes in the network, $V_P$ is the velocity penalty, $V_i$ is the flow velocity at link $i$, $T_V$ is the target velocity, $V_{P1}$ is the velocity penalty coefficient if the velocity at a given link is above target velocity, and $V_{P2}$ is the velocity penalty coefficient if the velocity at the link is below target velocity.

2.4.4. Sensitivity Analysis

Optimal solutions are not guaranteed in evolutionary algorithms; hence, maximizing 'near optimal' solutions is essential [64]. Genetic algorithm parameters are user input parameters that significantly affect the overall performance and speed of SOP-WDN. These parameters interact in a complex way [65] and must be tuned to obtain better solutions and faster convergence.

A sensitivity analysis was carried out to determine the most influential parameters and to obtain the best parameter combinations for effective execution of the algorithm. The parameters tested were:

1. Population Size
2. Crossover Probability
3. Mutation Rate
4. Velocity Penalty 1
5. Velocity Penalty 2

6. Pressure Penalty 1
7. Pressure Penalty 2

A total of 100 individual runs were performed (2000 iterations each) for any one alteration of any one of the seven parameters. Crossover rate and mutation rate were further tuned in combination since a proper balance between these operators is essential to ensure global optima. In total, 10–12 different values were tested in total for any one parameter and the value assigned per alteration was based on the literature and the preceding results. The final set of parameters selected, and the results obtained are given in Table 3 and Figure 5. It is to be noted that larger population size and iterations may be required for large water distribution networks.

**Table 3.** Set of Genetic Algorithm parameters.

| S.N. | GA Parameters | Values |
|:---:|:---:|:---:|
| 1 | Population Size | 80–100 |
| 2 | Crossover Probability (%) | 85–90 |
| 3 | Mutation Rate (%) | 4–6 |
| 4 | Velocity Penalty 1 | 0.3 |
| 5 | Velocity Penalty 2 | 0.06 |
| 6 | Pressure Penalty 1 | 0.02 |
| 7 | Pressure Penalty 2 | 1.9 |

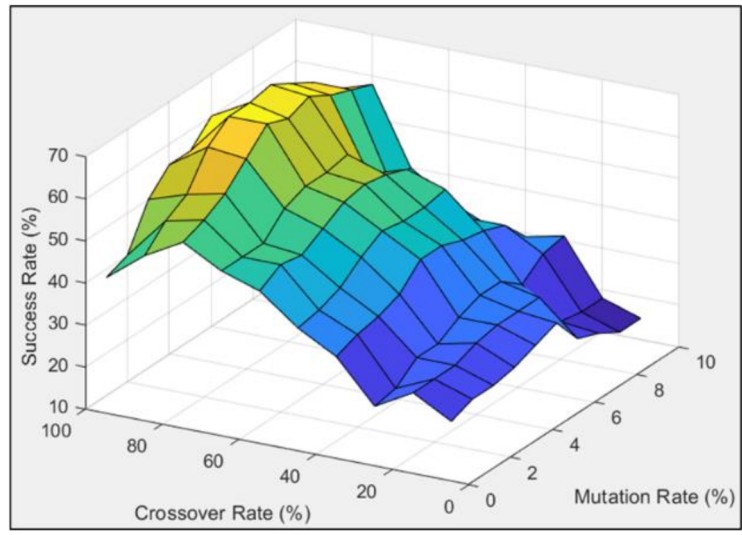

**Figure 5.** Sensitivity analysis results.

### 3. Results and Discussion

The conventional approach when testing the functionality, validity, and efficiency of a developed optimization algorithm is to choose some benchmark water distribution network problems and obtain their solution. Benchmark WDNs have provided a common testbed for newly developed optimization algorithms and design approaches. To prove their significance, the developed optimization algorithms are applied to benchmark WDN problems and are compared to the existing algorithms. Using SOP–WDN, some benchmark networks of the literature have been examined.

#### 3.1. Example 1: Two-Loop Network

The two-loop network is an imaginary network introduced by Alperovitz and Shamir [3] that consists of 8 pipelines and 7 nodes (with reservoir), all fed by gravity flow from a single reservoir with an elevation of 210 m. The layout of the two-loop network is given in Figure 6. All pipes in the layout are 1000 m in length, and the Hazen–Williams coefficient is 130. The minimum head requirement in each node is 30 m above ground level. The

commercially available diameters for the distribution network are described below on Table 4. The details of the distribution network have been provided in Appendix A:

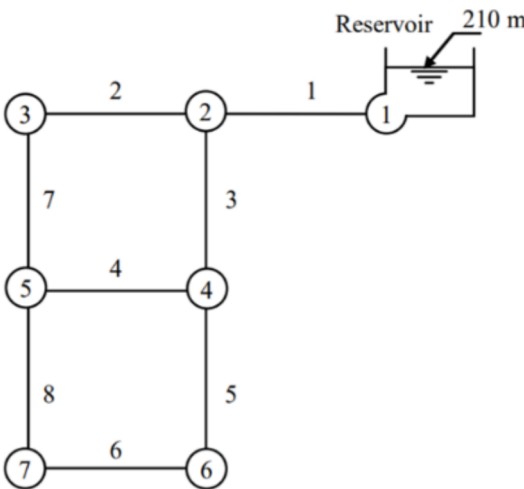

**Figure 6.** The Two-loop network layout.

**Table 4.** Available pipes for selection for the Two-loop network.

| Diameter (in) | Diameter (mm) | Unit Cost (USD/m) |
| --- | --- | --- |
| 1 | 25.4 | 2 |
| 2 | 50.8 | 5 |
| 4 | 101.6 | 11 |
| 6 | 152.4 | 16 |
| 10 | 254.0 | 32 |
| 14 | 355.6 | 60 |
| 16 | 406.4 | 90 |
| 18 | 457.2 | 130 |

Table 5 gives the solution obtained by SOP–WDN for the two-loop network. Figure 7 shows the EPANET network layout of the solved network and Figure 8 shows the pressure heads obtained at the nodes of the distribution network. Table 6 compares the solution obtained from SOP–WDN with the solution obtained by other research reports:

**Table 5.** SOP–WDN results for the Two-loop network.

| Pipe No. | Pipe Diameter (mm) | Pipe Length (m) | Cost (USD) | Node No. | Nodal Pressure (m) |
| --- | --- | --- | --- | --- | --- |
| 1 | 457.2 | 1000 | 130,000 | 1 | Reservoir |
| 2 | 254.0 | 1000 | 32,000 | 2 | 53.25 |
| 3 | 406.4 | 1000 | 90,000 | 3 | 30.46 |
| 4 | 101.6 | 1000 | 11,000 | 4 | 43.45 |
| 5 | 406.4 | 1000 | 90,000 | 5 | 33.81 |
| 6 | 254.0 | 1000 | 32,000 | 6 | 30.44 |
| 7 | 254.0 | 1000 | 32,000 | 7 | 30.55 |
| 8 | 25.4 | 1000 | 2000 | - | - |
| | Total Cost: | | 419,000 | Check | OK |

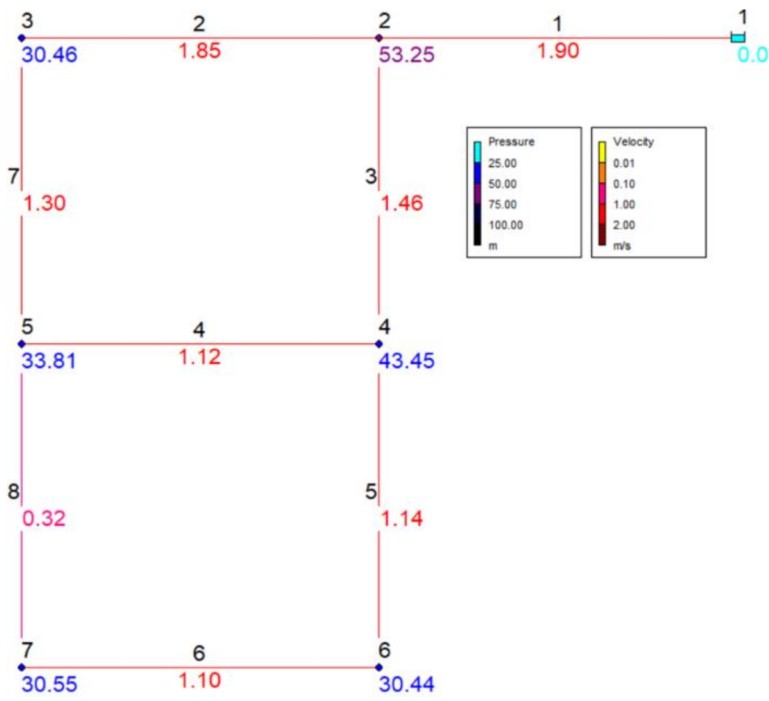

**Figure 7.** The Two-loop network solution obtained (pressure and velocity).

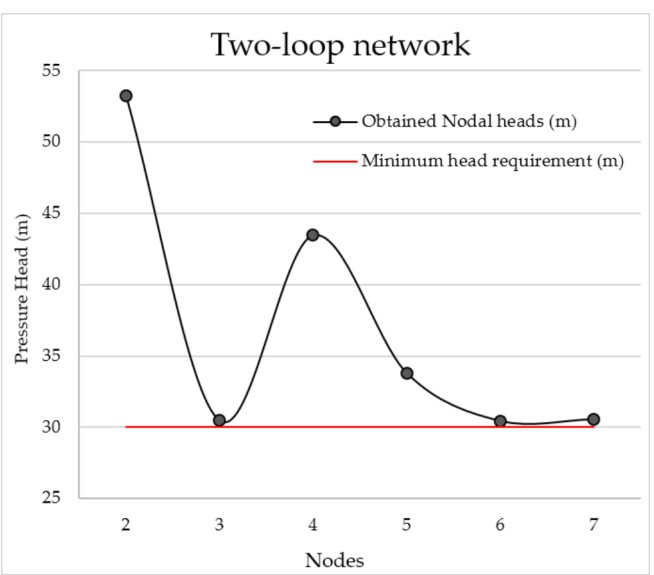

**Figure 8.** The Two-loop network nodal pressure heads.

**Table 6.** Comparison of SOP–WDN results to past studies for the Two-loop network.

| Studies | Alperovitz and Shamir | Savic and Walters | Geem | Van Dijk et al. | SOP–WDN |
| --- | --- | --- | --- | --- | --- |
| Least cost obtained (USD) | 479,525 | 420,000 | 419,000 | 419,000 | 419,000 |

The optimal cost of USD 419,000 was obtained for the Two-loop network, and the minimum pressure requirement of 30 m was fulfilled for all nodes. Table 6 shows the results obtained by other research reports for comparison. The optimal cost of USD 419,000 obtained by SOP-WDN for the two-loop network, is same as the solution obtained by Van Dijk et al. and Geem.

### 3.2. Example 2: Hanoi Network

Hanoi network, located in Vietnam, was first presented by Fujiwara and Kang [66]. It consists of 32 nodes, 34 pipes, and 3 loops, and is fed by gravity from a reservoir with a 100 m fixed head. The layout of the Hanoi network is given in Figure 9. All pipes available for this distribution network have a Hazen–Williams coefficient C of 130. The elevation of all nodes is 0 m, and minimum head limitation is 30 m above ground level. The commercially available diameters for the distribution network are described below on Table 7. The details of the distribution network have been provided in Appendix A.

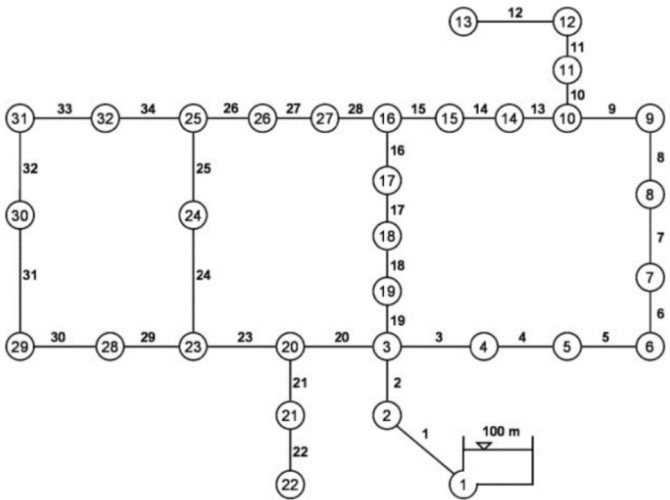

**Figure 9.** The Hanoi network layout.

**Table 7.** Available pipes for selection for the Hanoi network.

| Diameter (in) | Diameter (mm) | Unit Cost (USD/m) |
|---|---|---|
| 12 | 304.8 | 45.73 |
| 16 | 406.4 | 70.40 |
| 20 | 508 | 98.38 |
| 24 | 609.6 | 129.33 |
| 30 | 762 | 180.75 |
| 40 | 1016 | 278.28 |

Table 8 gives the solution obtained by SOP–WDN for the Hanoi network, while Table 9 compares the solution obtained from SOP–WDN with the solution obtained by other research reports. Figure 10 shows the EPANET network layout of the solved network and Figure 11 shows the pressure heads obtained at the nodes of the distribution network.

**Table 8.** SOP–WDN results for the Hanoi network.

| Pipe No. | Pipe Diameter (mm) | Pipe Length (m) | Cost (USD) | Node No. | Nodal Pressure (m) |
|---|---|---|---|---|---|
| 1 | 1016 | 100 | 27,828 | 1 | 100 (Reservoir) |
| 2 | 1016 | 1350 | 375,678 | 2 | 97.14 |
| 3 | 1016 | 900 | 250,452 | 3 | 61.67 |
| 4 | 1016 | 1150 | 320,022 | 4 | 56.92 |
| 5 | 1016 | 1450 | 403,506 | 5 | 51.02 |
| 6 | 1016 | 450 | 125,226 | 6 | 44.81 |
| 7 | 1016 | 850 | 236,538 | 7 | 43.35 |
| 8 | 1016 | 850 | 236,538 | 8 | 41.61 |
| 9 | 1016 | 800 | 222,624 | 9 | 40.23 |
| 10 | 762 | 950 | 171,712.5 | 10 | 39.20 |
| 11 | 609.6 | 1200 | 155,196 | 11 | 37.64 |
| 12 | 609.6 | 3500 | 452,655 | 12 | 34.21 |

**Table 8.** *Cont.*

| Pipe No. | Pipe Diameter (mm) | Pipe Length (m) | Cost (USD) | Node No. | Nodal Pressure (m) |
|---|---|---|---|---|---|
| 13 | 508 | 800 | 78,704 | 13 | 30.01 |
| 14 | 406.4 | 500 | 35,200 | 14 | 35.52 |
| 15 | 304.8 | 550 | 25,151.5 | 15 | 33.72 |
| 16 | 304.8 | 2730 | 124,842.9 | 16 | 31.30 |
| 17 | 406.4 | 1750 | 123,200 | 17 | 33.41 |
| 18 | 609.6 | 800 | 103,464 | 18 | 49.93 |
| 19 | 508 | 400 | 39,352 | 19 | 55.09 |
| 20 | 1016 | 2200 | 612,216 | 20 | 50.61 |
| 21 | 508 | 1500 | 147,570 | 21 | 41.26 |
| 22 | 304.8 | 500 | 22,865 | 22 | 36.10 |
| 23 | 1016 | 2650 | 737,442 | 23 | 44.52 |
| 24 | 762 | 1230 | 222,322.5 | 24 | 38.93 |
| 25 | 762 | 1300 | 234,975 | 25 | 35.34 |
| 26 | 508 | 850 | 83,623 | 26 | 31.70 |
| 27 | 304.8 | 300 | 13,719 | 27 | 30.76 |
| 28 | 304.8 | 750 | 34,297.5 | 28 | 38.94 |
| 29 | 406.4 | 1500 | 105,600 | 29 | 30.13 |
| 30 | 304.8 | 2000 | 91,460 | 30 | 30.42 |
| 31 | 304.8 | 1600 | 73,168 | 31 | 30.70 |
| 32 | 406.4 | 150 | 10,560 | 32 | 33.18 |
| 33 | 406.4 | 860 | 60,544 | - | - |
| 34 | 609.6 | 950 | 122,863.5 | - | - |
| | Total Cost: | | 6,081,115.4 | Check | OK |

**Table 9.** Comparison of SOP–WDN results to past studies for the Hanoi network.

| Studies | Savic and Walters | Liong and Atiquzzaman | Geem | Van Dijk et al. | SOP–WDN |
|---|---|---|---|---|---|
| Least cost obtained (Million USD) | 6.187 | 6.220 | 6.056 | 6.110 | 6.081 |

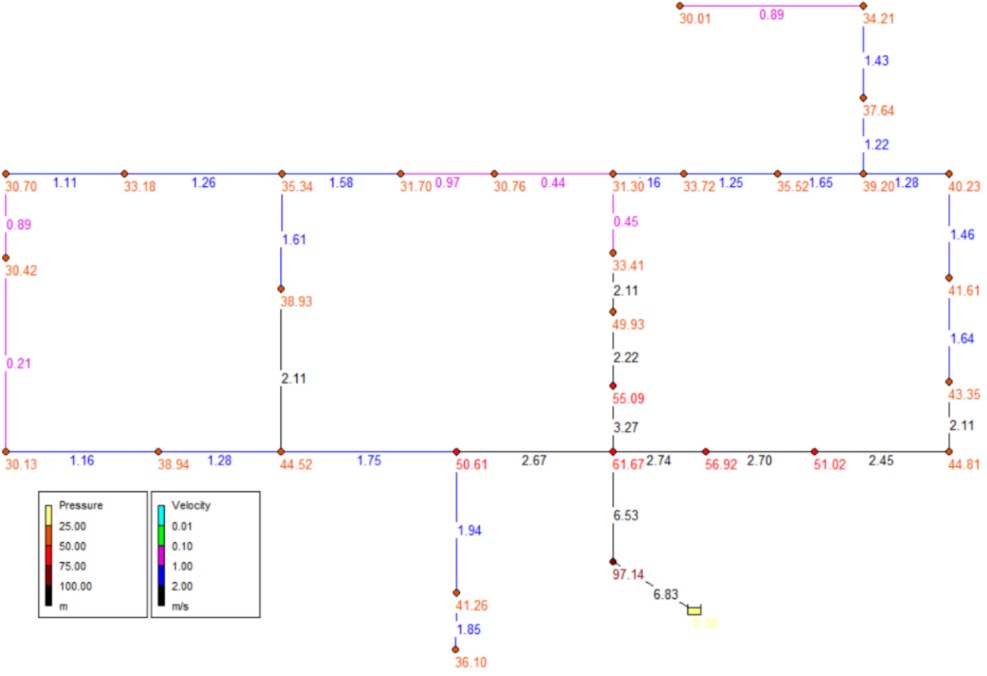

**Figure 10.** The Hanoi network solution obtained (pressure and velocity).

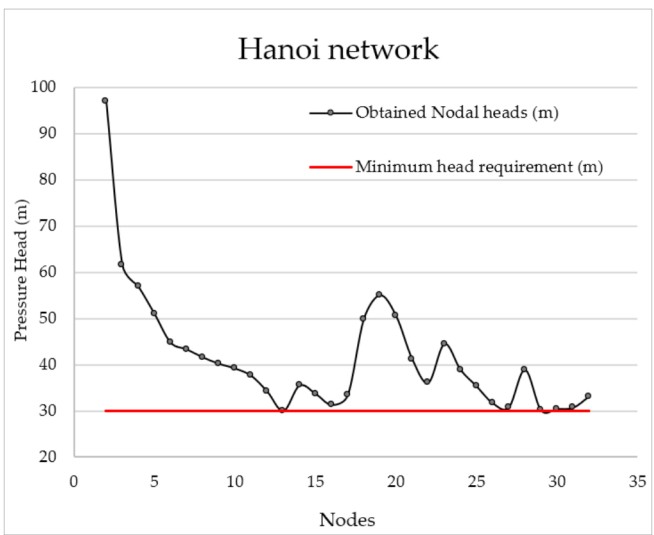

**Figure 11.** The Hanoi network nodal pressure heads.

The optimal cost of USD 6.081 million was obtained for Hanoi network, and the minimum pressure constraint of 30 m was fulfilled for all nodes. It was observed to be the best solution (lowest cost) without the violation of any constraints. The solution obtained by Geem [16] has a lower cost. However, this solution failed to meet the pressure constraint of 30 m at five nodes.

### 3.3. Example 3: GoYang Network

The GoYang water network is located in South Korea, and consists of 30 pipes, 22 nodes, and 9 loops. This network was first introduced by Kim et al. [67] and is fed by a single fixed pump of 4.52 kW from a 71 m constant head reservoir. The layout of the GoYang network is given in Figure 12. The Hazen–Williams coefficient for all pipes in the network is 100. The minimum head limitation for this network is 15 m above ground level. The commercially available diameters for the distribution network are described below on Table 10. The details of the distribution network have been provided in Appendix A.

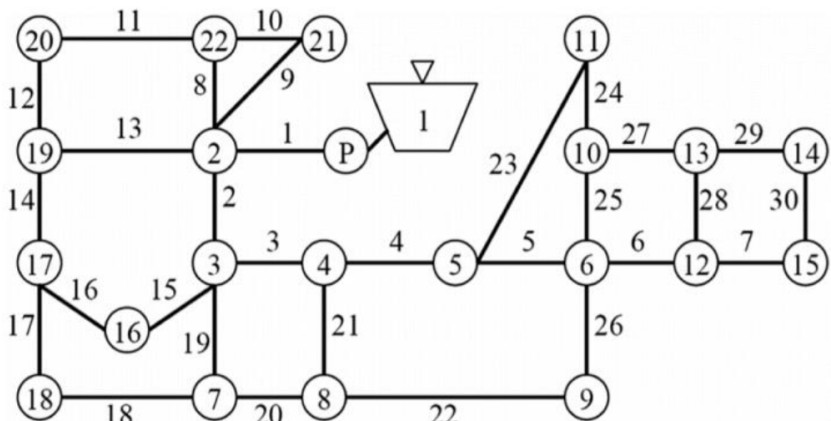

**Figure 12.** The GoYang network layout.

**Table 10.** Available pipes for selection for the GoYang network.

| Diameter (mm) | Unit Cost (Won/m) |
|---|---|
| 80 | 37,890 |
| 100 | 38,933 |
| 125 | 40,563 |
| 150 | 42,554 |
| 200 | 47,624 |
| 250 | 54,125 |
| 300 | 62,109 |
| 350 | 71,524 |

Table 11 gives the solution obtained by SOP–WDN for the GoYang network. Figure 13 shows the EPANET network layout of the solved network and Figure 14 shows the pressure heads obtained at the nodes of the distribution network. Table 12 compares the solution obtained from SOP–WDN with the solution obtained by other research reports.

**Table 11.** SOP–WDN results for the GoYang network.

| Pipe No. | Pipe Diameter (mm) | Pipe Length (m) | Cost (Won) | Node No. | Nodal Pressure (m) |
|---|---|---|---|---|---|
| 1 | 200 | 165.0 | 7,857,960 | 1 | 15.62 |
| 2 | 125 | 124.0 | 5,029,812 | 2 | 29.33 |
| 3 | 125 | 118.0 | 4,786,434 | 3 | 28.73 |
| 4 | 100 | 81.0 | 3,153,573 | 4 | 26.58 |
| 5 | 80 | 134.0 | 5,077,260 | 5 | 24.20 |
| 6 | 80 | 135.0 | 5,115,150 | 6 | 21.51 |
| 7 | 80 | 202.0 | 7,653,780 | 7 | 27.72 |
| 8 | 80 | 135.0 | 5,115,150 | 8 | 26.70 |
| 9 | 80 | 170.0 | 6,441,300 | 9 | 21.20 |
| 10 | 80 | 113.0 | 4,281,570 | 10 | 16.17 |
| 11 | 80 | 335.0 | 12,693,150 | 11 | 16.03 |
| 12 | 80 | 115.0 | 4,357,350 | 12 | 18.16 |
| 13 | 80 | 345.0 | 13,072,050 | 13 | 17.46 |
| 14 | 80 | 114.0 | 4,319,460 | 14 | 15.33 |
| 15 | 80 | 103.0 | 3,902,670 | 15 | 15.48 |
| 16 | 80 | 261.0 | 9,889,290 | 16 | 28.31 |
| 17 | 80 | 72.0 | 2,728,080 | 17 | 26.75 |
| 18 | 80 | 373.0 | 14,132,970 | 18 | 26.44 |
| 19 | 80 | 98.0 | 3,713,220 | 19 | 27.36 |
| 20 | 80 | 110.0 | 4,167,900 | 20 | 26.68 |
| 21 | 80 | 98.0 | 3,713,220 | 21 | 19.74 |
| 22 | 80 | 246.0 | 9,320,940 | 22 | 19.36 |
| 23 | 80 | 174.0 | 6,592,860 | - | - |
| 24 | 80 | 102.0 | 3,864,780 | - | - |
| 25 | 80 | 92.0 | 3,485,880 | - | - |
| 26 | 80 | 100.0 | 3,789,000 | - | - |
| 27 | 80 | 130.0 | 4,925,700 | - | - |
| 28 | 80 | 90.0 | 3,410,100 | - | - |
| 29 | 80 | 185.0 | 7,009,650 | - | - |
| 30 | 80 | 90.0 | 3,410,100 | - | - |
| | Total Cost: | | 177,010,359 | Check | OK |

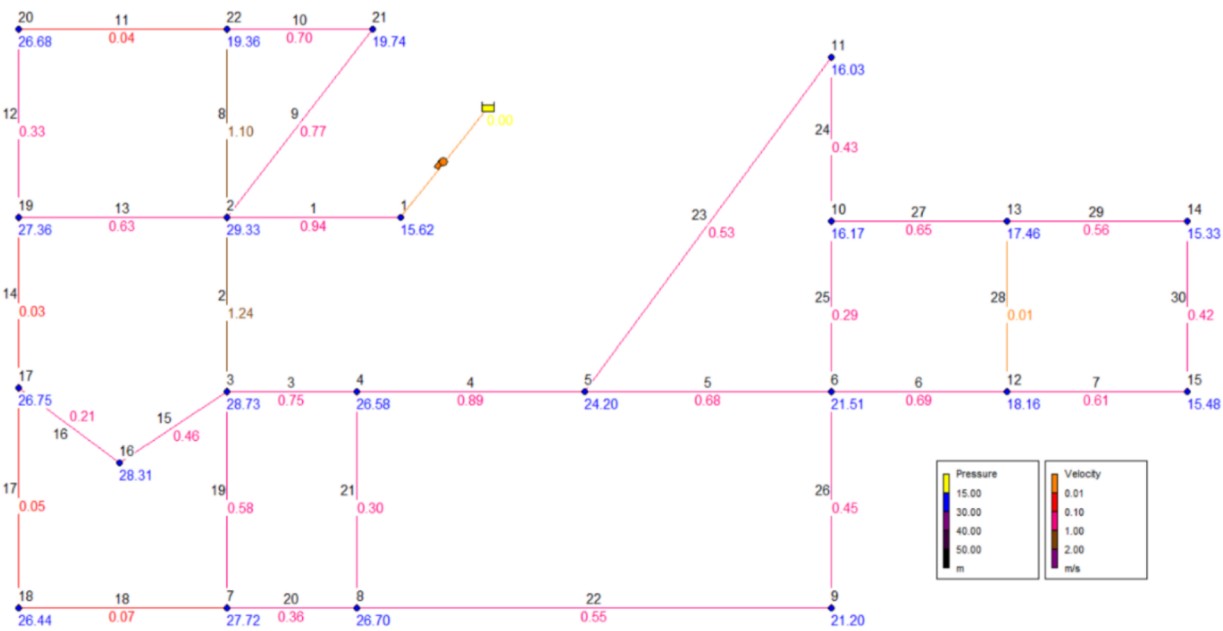

**Figure 13.** The GoYang network solution obtained (pressure and velocity).

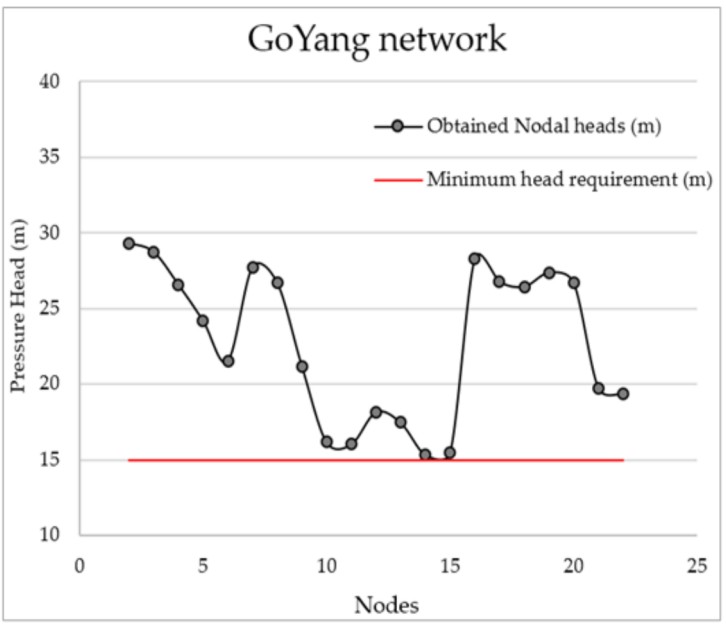

**Figure 14.** The GoYang network nodal pressure heads.

**Table 12.** Comparison of the SOP–WDN results to those of past studies of the GoYang network.

| Studies | Original Network | Kim et al. | Geem | Menon et al. [68] | SOP–WDN |
|---|---|---|---|---|---|
| Least cost obtained (Million Won) | 179.428 | 179.142 | 177.135 | 177.417 | 177.010 |

The lowest cost obtained by SOP–WDN for the GoYang network was 177,010,359 Won, which compared to the other studies, is the cheapest cost. The obtained solution also has no nodes containing pressure violations, as all nodes in the distribution network have fulfilled the minimum pressure requirement of 15 m.

## 4. Conclusions

In this study, the developed Genetic Algorithm based optimization algorithm SOP–WDN was tested on three benchmark water distribution networks, and in comparison, to the other studies, it was able to produce competitive results. EPANET software, which was used for the hydraulic analysis and calculations of the water distribution systems is a well-accepted and utilized software. EPANET–MATLAB toolkit enabled the SOP–WDN algorithm to perform EPANET based calculations directly in the MATLAB environment, which greatly improved the overall computational speed, performance, and efficiency of the algorithm. Hence, the EPANET–MATLAB toolkit can prove to be an important tool that enables the facile use of EPANET software in the MATLAB environment for many different research purposes. SOP–WDN can be used as a reliable algorithm and can easily be implemented and adapted to aid engineers and designers during the design process of new water distribution networks, or the rehabilitation of existing water distribution networks.

**Author Contributions:** U.S. developed SOP–WDN algorithm and wrote the manuscript of this study; K.-H.H., K.-M.K. and K.G. made recommendations during the writing of the manuscript; K.-T.Y. provided significant suggestions on the methodology and structure of the manuscript. All authors have read and agreed to the published version of the manuscript.

**Funding:** This work was supported by Korea Environment Industry & Technology Institute (KEITI) through Smart Water City Research Program of 2019002950004 and Development of Water and Sewage Innovation Technology Program of ARQ202001302001, funded by Korea Ministry of Environment (MOE).

**Institutional Review Board Statement:** Not applicable.

**Informed Consent Statement:** Not applicable.

**Data Availability Statement:** Datasets that are restricted and not publicly available.

**Conflicts of Interest:** The authors declare no conflict of interest.

## Appendix A

**Table A1.** Node data for the Two-loop network.

| Node No. | Elevation (m) | Demand (m³/h) |
|----------|---------------|----------------|
| 1 | 210 | Reservoir |
| 2 | 150 | 100 |
| 3 | 160 | 100 |
| 4 | 155 | 120 |
| 5 | 150 | 270 |
| 6 | 165 | 330 |
| 7 | 160 | 200 |

**Table A2.** Pipe data for the Two-loop network.

| Pipe No. | Begin Node | End Node | Length (m) |
|----------|------------|----------|------------|
| 1 | 1 | 2 | 1000 |
| 2 | 2 | 3 | 1000 |
| 3 | 2 | 4 | 1000 |
| 4 | 4 | 5 | 1000 |
| 5 | 4 | 6 | 1000 |
| 6 | 6 | 7 | 1000 |
| 7 | 3 | 5 | 1000 |
| 8 | 5 | 7 | 1000 |

**Table A3.** Node data for the Hanoi network.

| Node No. | Demand (m³/h) |
|---|---|
| 1 | Reservoir |
| 2 | 890 |
| 3 | 850 |
| 4 | 130 |
| 5 | 725 |
| 6 | 1005 |
| 7 | 1350 |
| 8 | 550 |
| 9 | 525 |
| 10 | 525 |
| 11 | 500 |
| 12 | 560 |
| 13 | 940 |
| 14 | 615 |
| 15 | 280 |
| 16 | 310 |
| 17 | 865 |
| 18 | 1345 |
| 19 | 60 |
| 20 | 1275 |
| 21 | 930 |
| 22 | 485 |
| 23 | 1045 |
| 24 | 820 |
| 25 | 170 |
| 26 | 900 |
| 27 | 370 |
| 28 | 290 |
| 29 | 360 |
| 30 | 360 |
| 31 | 105 |
| 32 | 805 |

**Table A4.** Pipe data for the Hanoi network.

| Pipe No. | Begin Node | End Node | Length (m) |
|---|---|---|---|
| 1 | 1 | 2 | 100 |
| 2 | 2 | 3 | 1350 |
| 3 | 3 | 4 | 900 |
| 4 | 4 | 5 | 1150 |
| 5 | 5 | 6 | 1450 |
| 6 | 6 | 7 | 450 |
| 7 | 7 | 8 | 850 |
| 8 | 8 | 9 | 850 |
| 9 | 9 | 10 | 800 |
| 10 | 10 | 11 | 950 |
| 11 | 11 | 12 | 1200 |
| 12 | 12 | 13 | 3500 |
| 13 | 10 | 14 | 800 |
| 14 | 14 | 15 | 500 |
| 15 | 15 | 16 | 550 |
| 16 | 17 | 16 | 2730 |
| 17 | 18 | 17 | 1750 |
| 18 | 19 | 18 | 800 |
| 19 | 3 | 19 | 400 |
| 20 | 3 | 20 | 2200 |

**Table A4.** *Cont.*

| Pipe No. | Begin Node | End Node | Length (m) |
|----------|-----------|----------|-----------|
| 21 | 20 | 21 | 1500 |
| 22 | 21 | 22 | 500 |
| 23 | 20 | 23 | 2650 |
| 24 | 23 | 24 | 1230 |
| 25 | 24 | 25 | 1300 |
| 26 | 26 | 25 | 850 |
| 27 | 27 | 26 | 300 |
| 28 | 16 | 27 | 750 |
| 29 | 23 | 28 | 1500 |
| 30 | 28 | 29 | 2000 |
| 31 | 29 | 30 | 1600 |
| 32 | 30 | 31 | 150 |
| 33 | 32 | 31 | 860 |
| 34 | 25 | 32 | 950 |

**Table A5.** Node data for the GoYang network.

| Node No. | Elevation (m) | Demand (m³/d) |
|----------|--------------|---------------|
| 1 | 71.0 | Reservoir |
| 2 | 56.4 | 153.0 |
| 3 | 53.8 | 70.5 |
| 4 | 54.9 | 58.5 |
| 5 | 56.0 | 75.0 |
| 6 | 57.0 | 67.5 |
| 7 | 53.9 | 63.0 |
| 8 | 54.5 | 48.0 |
| 9 | 57.9 | 42.0 |
| 10 | 62.1 | 30.0 |
| 11 | 62.8 | 42.0 |
| 12 | 58.6 | 37.5 |
| 13 | 59.3 | 37.5 |
| 14 | 59.8 | 63.0 |
| 15 | 59.2 | 445.5 |
| 16 | 53.6 | 108.0 |
| 17 | 54.8 | 79.5 |
| 18 | 55.1 | 55.5 |
| 19 | 54.2 | 118.5 |
| 20 | 54.5 | 124.5 |
| 21 | 62.9 | 31.5 |

**Table A6.** Pipe data for the GoYang network.

| Pipe No. | Begin Node | End Node | Length (m) |
|----------|-----------|----------|-----------|
| 1 | 1 | 2 | 165 |
| 2 | 2 | 3 | 124 |
| 3 | 3 | 4 | 118 |
| 4 | 4 | 5 | 81 |
| 5 | 5 | 6 | 134 |
| 6 | 6 | 12 | 135 |
| 7 | 12 | 15 | 202 |
| 8 | 2 | 22 | 135 |
| 9 | 2 | 21 | 170 |
| 10 | 21 | 22 | 113 |
| 11 | 22 | 20 | 335 |
| 12 | 20 | 19 | 115 |
| 13 | 2 | 19 | 345 |

**Table A6.** *Cont.*

| Pipe No. | Begin Node | End Node | Length (m) |
|---|---|---|---|
| 14 | 19 | 17 | 114 |
| 15 | 3 | 16 | 103 |
| 16 | 16 | 17 | 261 |
| 17 | 17 | 18 | 72 |
| 18 | 7 | 18 | 373 |
| 19 | 3 | 7 | 98 |
| 20 | 7 | 8 | 110 |
| 21 | 4 | 8 | 98 |
| 22 | 8 | 9 | 246 |
| 23 | 5 | 11 | 174 |
| 24 | 10 | 11 | 102 |
| 25 | 6 | 10 | 92 |
| 26 | 6 | 9 | 100 |
| 27 | 10 | 13 | 130 |
| 28 | 12 | 13 | 90 |
| 29 | 13 | 14 | 185 |
| 30 | 15 | 14 | 90 |

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
