# Peer review of "Optimization of Water Distribution Networks Using Genetic Algorithm Based SOP–WDN Program"

_water, doi:10.3390/w14060851_

Round 1
Reviewer 1 Report
The paper described the SOP-WDN computer program which calculates the optimum design of water distribution networks.
I find the results to be well documented and equal to or better than previous methods.
A few details which can be improved:
1) Line 99: "CT" should be "C_T".
2) Line 140: Explain "velocity defects".
3) Section 2.2: I find the discussion a bit unclear regarding open-source. I know that MATLAB requires a license, but is the EPANET-MATLAB toolkit free? I assume the toolkit still requires that you have a MATLAB license?
4) Line 233: Explain the "Hazen-Williams coefficient" or include a reference. Also, an abbreviated version (H-W) is used in line 283.
5) I think some of the larger tables should be placed in an Appendix.
Author Response
Thanks for the advice and pointing out what's wrong.
I have modified the part you mentioned and answered as follows.
Q1) Line 99: "CT" should be "C_T".
ïƒ The error in line 99 has been noted and has been corrected to CT. Please check Line: 101.
Q2) Line 140: Explain "velocity defects".
ïƒ The term velocity defect has been clarified and explained. Please check Line: 146.
Q3) Section 2.2: I find the discussion a bit unclear regarding open-source. I know that MATLAB requires a license, but is the EPANET-MATLAB toolkit free? I assume the toolkit still requires that you have a MATLAB license?
ïƒ Since, MATLAB itself does require license, the term open-source has been removed from the paper. Please check line: 186.
Q4) Line 233: Explain the "Hazen-Williams coefficient" or include a reference. Also, an abbreviated version (H-W) is used in line 283.
ïƒ Hazen-Williams coefficient has been explained in the paper including a reference for roughness coefficient values, please check: Line 107. All instances of (H-W) abbreviation has been removed.
Q5) I think some of the larger tables should be placed in an Appendix.
ïƒ The tables showing node data and pipe data have been moved to Appendix A.
Reviewer 2 Report
The authors address the problem of water distribution networks and provide a new heuristic algorithm to deal with.
- Problem formulation should be moved in section 2, before introducing the genetic algorithm.
- Section 1: Introduction and related work section should be two different sections.
- "EPANET is a computer program" -> I think the term algorithm is more pertinent with respect to the term computer program
- Abstract: "computer program" -> "algorithm" (I suggest replacing it in all sections of the paper)
- "Before running the program, the network layout and network data must be imported as an .INP 195 file from EPANET." The authors provide many technical details that could be removed.
- The discussion on the GA algorithms in section 2.3 should be supported with a concrete example. I suggest showing a small pipe network aiming to illustrate different steps of the GA.
- A more detailed discussion on the Selection, Crossover, Mutation, and Elitism is required.
- It is not clear the strategies adopted for Selection, Crossover, Mutation, and Elitism. I suggest adding a subsection for each step in which authors further detail these steps, also introducing an example on them.
- "Section 3.2. Example 2: Hanoi network" -> The section should show an experimental evaluation, not examples.
- Although the authors present a flowchart, pseudocode of the algorithm is required.
Author Response
Thanks for the advice and pointing out what's wrong.
I have modified the part you mentioned and answered as follows.
Q1) Problem formulation should be moved in section 2, before introducing the genetic algorithm.
ïƒ Problem formulation has been moved into section 2 before introduction of Genetic Algorithm.
Q2) Section 1: Introduction and related work section should be two different sections.
ïƒ The section has been divided into two sections (introduction and background).
Q3) "EPANET is a computer program" -> I think the term algorithm is more pertinent with respect to the term computer program
ïƒ The term has been replaced as such “EPANET is a hydraulic simulator”.
Q4) Abstract: "computer program" -> "algorithm" (I suggest replacing it in all sections of the paper)
ïƒ All sections of the paper containing “computer program” has been replaced with “algorithm”.
Q5) "Before running the program, the network layout and network data must be imported as an .INP 195 file from EPANET." The authors provide many technical details that could be removed.
ïƒ Such technical details have either been reworded or removed entirely. Please check line: 201.
Q6) The discussion on the GA algorithms in section 2.3 should be supported with a concrete example. I suggest showing a small pipe network aiming to illustrate different steps of the GA.
ïƒ An example has been illustrated on section 2.4.1.
Q7) A more detailed discussion on the Selection, Crossover, Mutation, and Elitism is required.
ïƒ Genetic Operators have been briefly explained on section 2.4.2.
Q8) It is not clear the strategies adopted for Selection, Crossover, Mutation, and Elitism. I suggest adding a subsection for each step in which authors further detail these steps, also introducing an example on them.
ïƒ Genetic Operators have been briefly explained on section 2.4.2 and 2.4.4.
Q9) "Section 3.2. Example 2: Hanoi network" -> The section should show an experimental evaluation, not examples.
ïƒ Hanoi network is a benchmark network utilized by many researchers to validate the functionality of their optimization algorithm. Hence, it was used to demonstrate the results obtained by SOP-WDN and compare it to the results obtained by other studies.
Q10) Although the authors present a flowchart, pseudocode of the algorithm is required.
ïƒ In section 2.4 the authors explain the flowchart, interpretation by the algorithm, genetic operators and sensitivity analysis to explain the workings of the algorithm. Unfortunately, pseudocode of the algorithm could not be provided.
Reviewer 3 Report
Manuscript ID: water-1595446
Title: Optimization of Water Distribution Networks Using Genetic Algorithm Based SOP–WDN Program.
OVERVIEW
The manuscript presents the program SOP–WDN and demonstrates the application of an evolutionary optimization technique, Genetic Algorithm, linked with a hydraulic simulation solver EPANET, for the optimal design of water distribution networks.
- The subject matter is actual, interesting and within the scope of the Journal Water.
- The title fully describes the manuscript.
- The manuscript complies with the journal template and is well structured.
- The introduction gives an insight into the optimal design of water distribution networks and presents relevant bibliographical references.
- The results presented by the authors are exciting as the developed algorithm can obtain the best solutions in the benchmark WDN problems.
- The English is good.
- As for the rest, I have a few suggestions. Please read the specific comment.
In conclusion, I believe this manuscript is worthy of publication, after a minor revision.
SPECIFIC COMMENTS
Please describe in the manuscript the applied GA operators for selection, crossover, mutation and elitism.
The optimum solution for the benchmark WDN problems is presented but it is not provided information about the number of iterations and computational time required to obtain the optimum solution. Please include that information in the manuscript and rate the computational efficiency of the algorithm.
Author Response
Thanks for the advice and pointing out what's wrong.
I have modified the part you mentioned and answered as follows.
Q1) Please describe in the manuscript the applied GA operators for selection, crossover, mutation and elitism.
ïƒ Two new sections have been added in the text – GA Operators (2.4.2) and Sensitivity Analysis (Section 2.4.4) that explain about the GA operators used in the algorithm.
Q2) The optimum solution for the benchmark WDN problems is presented but it is not provided information about the number of iterations and computational time required to obtain the optimum solution. Please include that information in the manuscript and rate the computational efficiency of the algorithm.
ïƒ Since the time required to obtain the solution largely depends upon the power of the computer processor. It has not been included. However, the iterations and computational efficiency of the program have been discussed in section (2.4.4)
Round 2
Reviewer 2 Report
Thanks to the authors for following the previous comments. The overall quality of the paper has been improved. However, there are some minor comments to be reviewed:
- The quality of Figure 1 should be improved.
- Selection step is a crucial phase in GA algorithms. It is not clear how authors determine the best candidates to be selected.
- Related works are few and many are old. In addition, other research areas also face similar optimization problems. I would like to suggest some related works with which authors can develop the related works section: https://doi.org/10.1016/j.oceaneng.2021.110261, http://ceur-ws.org/Vol-2037/paper_22.pdf, https://doi.org/10.1109/iV.2018.00030, https://ascelibrary.org/doi/abs/10.1061/(ASCE)0733-9496(2004)130:1(73)
- It is not clear why the authors choose to select only the best candidates. I might agree with the fact that we need good candidates in the population, but this could make the algorithm converge into a local optimal rather than a global one. I suggest further discussing and clarifying this.
- Figure 1 shows that the Fitness evaluation is performed before the GA operations. However, in Section 2.4.2. authors state that the fitness of the individuals is evaluated in the selection step.
- The contents of Sections 2, 3, and 4 can be merged.
- The approaches adopted in the crossover and mutation steps are well-known in the literature. This could represent a limitation for the novelty of the GA. I suggest comparing the effectiveness of other strategies to further demonstrate the effectiveness of the proposed approaches.
- Crossover and mutation are poorly discussed and motivated. Further discussion on them is required.
- It is not clear how the authors manage mutation and crossover probabilities. Do populations always mutate? Is there always crossover on individuals? This does not match the criteria of genetic algorithms in which individuals mutate and cross with a certain probability.
The paper requires further clarification on the methodology and steps underlying the GA. Furthermore, authors should further emphasize the novelty with respect to the state of the art.
Author Response
Thank you very much.
The introduction section has been modified and improved. Newer publications relevant to the topic of the article have been added as literature review. The results section has also been improved, the results obtained have been added on the table and clearly explained.
Q1) The quality of Figure 1 should be improved.
-> Figure 1 has been improved and the workings have been explained more clearly.
Q2) Selection step is a crucial phase in GA algorithms. It is not clear how authors determine the best candidates to be selected.
-> Yes, the selection step is one of the most vital steps in the GA that must be considered carefully for obtaining global optima and for improving the speed of convergence. The selection operator has been more clearly explained.
Q3) Related works are few and many are old. In addition, other research areas also face similar optimization problems. I would like to suggest some related works with which authors can develop the related works section: https://doi.org/10.1016/j.oceaneng.2021.110261, http://ceur-ws.org/Vol-2037/paper_22.pdf, https://doi.org/10.1109/iV.2018.00030, https://ascelibrary.org/doi/abs/10.1061/(ASCE)0733-9496(2004)130:1(73)
-> We would like to sincerely thank the reviewer for their suggestions since it has greatly improved the quality of the article. Newer publications relevant to the topic of the article and some related works have been added.
Q4) It is not clear why the authors choose to select only the best candidates. I might agree with the fact that we need good candidates in the population, but this could make the algorithm converge into a local optimal rather than a global one. I suggest further discussing and clarifying this.
-> The authors would like to apologize for poor explanation of candidate selection. Only the best candidate is Not chosen from the population (since it would definitely converge into a local optima). The candidates are ranked among based on their fitness and then randomly selected. Hence, the fitter candidates have higher chance of being selected. However, the poor candidates still have some change. Due to this genetic information of poor candidates (which might be valuable genetic information) get passed. This has been explained in a better way.
Q5) Figure 1 shows that the Fitness evaluation is performed before the GA operations. However, in Section 2.4.2. authors state that the fitness of the individuals is evaluated in the selection step.
-> This error in explanation has been noted and changed. Figure 1 has been updated and Selection operator has been explained more clearly on section 2.4.2.
Q6) The contents of Sections 2, 3, and 4 can be merged.
-> Since, lots of new information have been added into the article based on reviewer’s and editor’s comments. The contents have been left as they were.
Q7) The approaches adopted in the crossover and mutation steps are well-known in the literature. This could represent a limitation for the novelty of the GA. I suggest comparing the effectiveness of other strategies to further demonstrate the effectiveness of the proposed approaches.
-> The strategies used that resulted in improving the GA have been better explained in the article.
Q8) Crossover and mutation are poorly discussed and motivated. Further discussion on them is required.
-> Crossover and mutation operations have been explained in greater detail.
Q9) It is not clear how the authors manage mutation and crossover probabilities. Do populations always mutate? Is there always crossover on individuals? This does not match the criteria of genetic algorithms in which individuals mutate and cross with a certain probability.
-> No, there is not always crossover on individuals. The crossover probability of 85%-90% and mutation probability of 4-6% was found to produce best results and were chosen as such. These issues have been properly discussed on the article.
Q10) The paper requires further clarification on the methodology and steps underlying the GA. Furthermore, authors should further emphasize the novelty with respect to the state of the art.
-> The internal workings of the algorithm have been explained in more detail and some problems (that are not well explored in literature) that occur when applying Binary coded GA for WDN optimization have been mentioned and how they are tackled have also been explained.
Round 3
Reviewer 2 Report
First of all, I would like to thank the authors for following previous remarks and improving the overall quality of the paper.
- The quality of Figure 1 is now improved, and the errors have been solved.
- Previous remarks: I would like to suggest some related works with which authors can extend the related works section: https://doi.org/10.1016/j.oceaneng.2021.110261, http://ceur-ws.org/Vol-2037/paper_22.pdf, https://doi.org/10.1109/iV.2018.00030, https://ascelibrary.org/doi/abs/10.1061/(ASCE)0733-9496(2004)130:1(73)
- Although Crossover and mutation have been further discussed, it is not clear why the authors did not have designed specific crossover and mutation strategies for this problem.
- Results: I would suggest drawing graphs to summarize the table in the results section. All tables can be easily represented as graphs to improve readability.
Author Response
Thanks for the good advice.
Q1) The quality of Figure 1 is now improved, and the errors have been solved.
-> Yes, the authors agree. We would like to thank the Reviewer for their suggestions regarding improvement of the Figure.Q
Q2) Previous remarks: I would like to suggest some related works with which authors can extend the related works section: https://doi.org/10.1016/j.oceaneng.2021.110261, http://ceur-ws.org/Vol-2037/paper_22.pdf, https://doi.org/10.1109/iV.2018.00030, https://ascelibrary.org/doi/abs/10.1061/(ASCE)0733-9496(2004)130:1(73)
-> This related work was already added during Round-2, into the article’s related work section: https://ascelibrary.org/doi/abs/10.1061/(ASCE)0733-9496(2004)130:1(73). Ref [20]
Also, the other related works suggested by the reviewer, have now been added in the related work section. Ref [6], Ref [48,49].
Q3) Although Crossover and mutation have been further discussed, it is not clear why the authors did not have designed specific crossover and mutation strategies for this problem.
-> The comment has been further addressed in the article.
Q4) Since, in WDN optimization using GA, unlike any traditional function optimization, the change of any single gene in a chromosome means changing and replacing the current pipe to another one of a different diameter. Hence, the methods described in the article: utilizing K-point with 0.8×Np points for crossover and Bit-flip with hill-climbing for the final 10% of generation (mainly to improve convergence) for mutation produced good results. Furthermore, specific strategies in other section may overshadow the method utilized for redundancy handling.- Results: I would suggest drawing graphs to summarize the table in the results section. All tables can be easily represented as graphs to improve readability.
-> Graphs have been added in the result section for easy representation of the results obtained.